# Aging Activates the Immune System and Alters the Regenerative Capacity in the Zebrafish Heart

**DOI:** 10.3390/cells11030345

**Published:** 2022-01-20

**Authors:** Hanna Reuter, Birgit Perner, Florian Wahl, Luise Rohde, Philipp Koch, Marco Groth, Katrin Buder, Christoph Englert

**Affiliations:** 1Molecular Genetics Laboratory, Leibniz Institute on Aging—Fritz Lipmann Institute (FLI), 07745 Jena, Germany; Hanna.reuter@leibniz-fli.de (H.R.); Birgit.perner@leibniz-fli.de (B.P.); Florian.wahl@uni-jena.de (F.W.); Luise.rohde@uni-jena.de (L.R.); 2Core Facility Imaging, 07745 Jena, Germany; 3Core Facility Life Science Computing, 07735 Jena, Germany; Philipp.Koch@leibniz-fli.de; 4Core Facility DNA Sequencing, 07745 Jena, Germany; Marco.Groth@leibniz-fli.de; 5Core Service Histology/Pathology/EM, Leibniz Institute on Aging—Fritz Lipmann Institute (FLI), 07745 Jena, Germany; Kartin.buder@leibniz-fli.de; 6Institute of Biochemistry and Biophysics, Friedrich-Schiller-University Jena, 07745 Jena, Germany

**Keywords:** aging, zebrafish heart, age-dependent regeneration, immune cells, cryoinjury

## Abstract

Age-associated organ failure and degenerative diseases have a major impact on human health. Cardiovascular dysfunction has an increasing prevalence with age and is one of the leading causes of death. In contrast to humans, zebrafish have extraordinary regeneration capacities of complex organs including the heart. In addition, zebrafish has recently become a model organism in research on aging. Here, we have compared the ventricular transcriptome as well as the regenerative capacity after cryoinjury of old and young zebrafish hearts. We identified the immune system as activated in old ventricles and found muscle organization to deteriorate upon aging. Our data show an accumulation of immune cells, mostly macrophages, in the old zebrafish ventricle. Those immune cells not only increased in numbers but also showed morphological and behavioral changes with age. Our data further suggest that the regenerative response to cardiac injury is generally impaired and much more variable in old fish. Collagen in the wound area was already significantly enriched in old fish at 7 days post injury. Taken together, these data indicate an ‘inflammaging’-like process in the zebrafish heart and suggest a change in regenerative response in the old.

## 1. Introduction

The gradual decline of tissue functionality is the main reason humans suffer from age-related diseases. The ability to replace missing or damaged tissue, organs or even entire body parts and functionally integrate these regenerates into the existing structures is limited to a few organs, including the liver, intestine, skeletal muscle and the hematopoietic system. Organ failure and degenerative diseases thus have a major impact on human health, especially when we age. Cardiovascular diseases are the leading cause of death, representing 32% of all deaths globally in 2019 (World Health Organization—www.who.int (accessed on 3 September 2021)). The prevalence of cardiovascular diseases escalates with increasing age, that is, a cross-sectional study reports for England such diseases to be present in less than 1% of people under the age of 50 but in over 40% of people over the age of 80 years [1]. A considerable part of this morbidity and mortality is due to irreversible damage of the heart muscle. Myocardial infarction, the acute blockage of a coronary artery, leads to the formation of a non-contractile scar post injury that impairs heart function. Hence, prevention and treatments to improve patient care are needed, especially in the elderly. In order to prevent age-related cardiac diseases, it is therefore of importance to understand the respective age-associated risk factors. In addition, it is of great interest to understand molecular mechanisms underlying regeneration processes to devise and improve regenerative strategies in humans.

The potential to regenerate tissues and organs is widespread in the animal kingdom; however, it is extremely variable. In contrast to humans, other vertebrates such as the zebrafish have extraordinary regeneration capacities to repair complex organs such as the telencephalon, spinal cord, kidney and appendages after severe injury [2,3,4,5]. The zebrafish heart has been proven to possess robust regenerative capacities and several injury models are available, namely ventricular apex amputation, cryoinjury, genetic cardiomyocyte depletion, and hypoxia [6,7,8,9,10,11]. The cryoinjury method induces local necrosis, triggers apoptosis and involves the formation of a fibrotic scar. That is why this method is considered a model that resembles myocardial infarction in humans. The zebrafish heart consists of a single atrium and a single ventricle with blood entering via the sinus venosus and exiting through the bulbus arteriosus, which is made of smooth muscle and non-contractile. The epicardium covers the myocardium externally and the endocardium is the innermost layer. The zebrafish cardiac muscle can be divided into the cortical myocardium (compact layer) that is supported by coronary vessels and the inner trabecular myocardium with sponge-like muscular structures. Several key steps in the regeneration process of the heart have been identified for zebrafish (reviewed in [12]). During the early response to ventricular cryoinjury, tissue death triggers an inflammatory response that involves the recruitment of immune cells to the wound area. Epicardial and endocardial cells get activated, proliferate and cover the injured area. Myofibroblasts appear in the injured area and accumulation of extracellular matrix occurs, resulting in the formation of a fibrotic scar. Cardiomyocytes at the border zone between wound and intact muscle proliferate and repopulate the injury zone. The fibrotic tissue is eventually efficiently resolved. In zebrafish, cardiac fibrosis is thus only observed transiently, in contrast to humans. It has been shown that the heart not only regenerates on a structural level but the regenerated myocardium also becomes electrically coupled and electrical activity of the heart returns to normal [8,13]. Only the regenerated muscle wall shows an expansion of the cortical myocardium after the regeneration process is completed. On the contrary, the heart in mammals is considered a non-regenerative organ harboring post-mitotic and terminally differentiated tissue. However, studies using stable isotope incorporation have provided evidence for cardiomyocyte renewal in humans, at very low rates [14,15]. Although the rare events of cardiomyocyte renewal cannot compensate for the damage of a myocardial infarction, these results lead to hope that therapeutic strategies that help driving myocardial renewal might eventually be developed even for humans.

An aging related decline in the regenerative potential of specific tissues in mammals has been observed, that is, for skeletal muscle and the liver [16,17]. Similarly, age-dependent decline in the capacity to regenerate the caudal fins has been reported for the short-lived fish *Nothobranchius furzeri*, suggesting that regenerative processes are initiated later in the old and that overall regeneration is more efficient in younger fish [18]. Interestingly, the neonatal mouse hearts possesses regenerative potential that is lost by 7 days of age [19]. These findings indicate that regeneration potential significantly depends on the age of the organism. In contrast, life-long regeneration has been proposed for the zebrafish fin as well as for the heart [20]. Other studies reported that age-related declined regenerative capacity correlates with impaired fin formation [21,22]. These differences might result from a rather wide or non-uniform definition of “old” in zebrafish. Given a median life span of approximately 36 to 42 months and a maximum life span of up to 66 months [23], fish used in the study reporting life-long capacities were 26- to 36-month-old and might not represent a cohort of truly old fish.

After decades of developmental research, zebrafish has recently also become a model organism in research on aging. The zebrafish shows several changes with age that are typical signs of aging in humans, on the cellular and molecular as well as the organismic level. Such phenotypes include the appearance of senescence-associated beta-galactosidase activity in the skin and the accumulation of oxidized proteins in the muscle, as well as spinal curvature and skeletal degeneration [24,25]. Furthermore, decreased neurogenesis has been reported for the brain as well as impairment of cognitive function [26,27,28]. Concerning the zebrafish heart, age-associated changes in electrical function, pathophysiological changes and an increase of sinus arrest episodes have been described [29,30,31]. Additionally, in recent studies, the use of genetic manipulations in zebrafish has helped in defining and investigating molecular drivers of aging or longevity, such as *celsr1a*, *klotho* and *rag1*, encoding a non-classical cadherin, a hormone and a lymphoid-specific endonuclease, respectively [32,33,34].

Here, we have compared the ventricular transcriptome as well as the regenerative capacity after cryoinjury of 7-month-young and 4-year-old zebrafish hearts. We identify the immune system as increasingly activated with age and show an accumulation of immune cells, mostly macrophages, in the old zebrafish ventricle. Our data further suggest that the regenerative response to cardiac injury is generally impaired and much more variable in old fish with a subset of animals still showing young-like regenerative capacity at an age of more than 3.5 years.

## 2. Materials and Methods

### 2.1. Fish Maintenance

Wild type zebrafish (*Danio rerio*) were obtained from matings of the AB and TU strains. All fish were housed in the fish facility at the Leibniz Institute on Aging—Fritz Lipmann Institute in standard conditions under a 14-h-light and 10-h-dark cycle. All animal procedures were performed in accordance with the rules of the German Animal Welfare Law and approved by the Landesamt für Verbraucherschutz Thüringen, Germany. Published strains used in this study include: wild-type AB TU, *Tg(pu.1:GFP)*^zdf11Tg^ [35] and *Tg(mpeg1.1:NTR-EYFP)*^w202Tg^ [36].

### 2.2. Zebrafish Cardiac Injuries and Sample Collection

Cardiac ventricular cryoinjury experiments were conducted using adult zebrafish as described previously [37]. Adult fish were anesthetized, their pericardial cavity opened and a copper filament pre-cooled in liquid nitrogen was placed on the ventricular surface of the exposed heart until thawing. Pericardial cavity of sham treated animals was opened but no heart damage was induced. Adult zebrafish were euthanized with an overdose of tricaine and hearts dissected out and processed as described previously [37]. Similarly, hearts of uninjured young and old fish were collected. Sample sizes and age are indicated in each figure legend.

### 2.3. RNA-Sequencing

Total RNA was extracted from zebrafish ventricles using trizol according to the manufacturer’s protocol. Sequencing of RNA samples was performed using Illumina’s next-generation sequencing methodology [38]. In detail, 11 total RNA samples were quantified and quality checked using Agilent 2100 Instrument in combination with RNA 6000 Pico assay (both Agilent Technologies, Santa Clara, CA, USA). Three samples per group (young and old) were chosen based on the RIN (RNA Integrity Number) in order to guarantee using samples with the best quality. Libraries were prepared from 160 ng of input material (total RNA) using NEBNext Ultra II Directional RNA Library Preparation Kit in combination with NEBNext Poly(A) mRNA Magnetic Isolation Module following the manufacturer’s instructions (New England Biolabs, Ipswich, MA, USA). Quantification and quality check of libraries was done using an Agilent 2100 Bioanalyzer Instrument and DNA 7500 assay (Agilent Technologies, Santa Clara, CA, USA). Libraries were pooled and sequenced using a HiSeq 2500 system (San Diego, CA, USA) in 51 cycle/single-end/high-output mode. Sequence information was converted to FASTQ format using bcl2fastq v2.20.0.422.

### 2.4. Differential Gene Expression Analysis

The RNA sequencing reads were aligned to the zebrafish reference genome (GRCz11 with the Ensembl genome annotation Release 95) using STAR 2.5.4b (parameters: --alignIntronMax 100,000, --outSJfilterReads Unique, --outSAMmultNmax 1, --outFilterMismatchNoverLmax 0.04) [39]. For each gene, all reads that mapped uniquely to one genomic position were counted with FeatureCounts 1.6.3 (multi-mapping or multi-overlapping reads were discarded, stranded mode was set to “–s 2”) [40]. Differentially expressed genes (DEGs) were determined with R 3.5.2 using the package DESeq2 1.22.2 [41]. Only genes that have at least one read count in any of the analyzed samples of a particular comparison were subjected to DESeq2. Three old zebrafish ventricle samples were contrasted to three young ventricle samples. For each gene in each comparison, the *p*-value was calculated using the Wald significance test. Resulting *p*-values were adjusted for multiple testing using the Benjamini and Hochberg correction. The log2 fold change values were shrunk with the DESeq2 function lfcShrink (type = “normal”) to control for variance of L2FC estimates for genes with low read counts. Genes with an adjusted *p* < 0.05 were considered differentially expressed.

### 2.5. Gene Ontology Enrichment Analysis

Over-represented GO terms of the differentially up or down regulated genes (adjusted *p* value < 0.05) were determined with R using the Clusterprofile package [42] and itsenrichGO function (with pAdjustMethod = “BH”, pvalueCutoff = 0.01, qvalueCutoff = 0.05). Results were visualized with *dotplot* or *cnetplot* (with circular = TRUE).

### 2.6. Data Comparison with Klotho Mutant

We compared the Ensembl IDs of our 1233 DEGs in old vs. young ventricles with the Ensembl IDs of 4927 genes identified as differentially expressed in hearts of *klotho* mutants compared to wild types [32] using R. Results were visualized with the package *VennDiagram* and *ggplot2*. Over-represented GO terms of the overlapping genes were determined as described above.

### 2.7. Electron Microscopy

Heart ventricles of three young and three old fish were isolated and immediately fixed in Karnovsky Fixative (4% PFA/5% Glutaraldehyde in 0.1 M cacodylate buffer, pH 7.3) for at least one day at 4 °C. For a secondary fixation the samples were incubated in 2% OsO_4_/1% Potassium ferrocyanide in 0.1 M cacodylate buffer for 3 h at 4 °C in the dark followed by dehydration in an ascending water/acetone series and embedded in epoxy resin ‘Epon’ (glycid ether 100, SERVA, Heidelberg, Germany). The resin was allowed to polymerize for 2 d at 60 °C in flat embedding molds. After curing the samples were trimmed with a Reichert UltraTrim (Leica, Wetzlar, Germany). Sagittal semithin sections of 0.5 µm with an Azure staining (Richardson et al., 1960) were executed for light microscopy analysis. Ultrathin sections of 55 nm without poststaining were placed onto copper slot grids coated with a Formvar/Carbon layer for TEM analysis. All sections were made with an ultramicrotome (Reichert Ultracut S; Leica, Wetzlar, Germany) and electron micrographs were taken on a JEM 1400 electron microscope (JEOL, Akishima, Japan) using an accelerating voltage of 80 kV and coupled with Orius SC 1000 CCD-camera (GATAN, Pleasanton, CA, USA). Software: GATAN MICROSCOPY SUITE Vers. 2.31.734.0.

### 2.8. Histological and Immunofluorescence Staining of Sections

Hearts were fixed in 4% PFA overnight, either dehydrated with a series of 4%, 20% and 30% sucrose before freezing in cryo-embedding medium (Neg-50—Thermo Scientific, Waltham, MA, USA) or dehydrated with an increasing gradient of ethanol before xylene and paraffin treatment using an automated vacuum tissue processor (Thermo Fisher Scientific, Waltham, MA, USA). Serial tissue sections were collected from paraffin and cryo embedded samples as described in [43] using a microtome or microm cryostat, respectively. Acid fuchsin-orange G (AFOG) stain was used on paraffin tissue sections to detect fibrotic tissue (muscle brown-orange, fibrin/cell debris red, collagen blue) [6]. Deparaffination was carried out using an automated slide stainer (Leica, Wetzlar, Germany). In situ hybridizations were performed on paraffin sections and Oil red O stain as well as SA-activated beta-galactosidase stainings were performed on cryo tissue sections to detect lipid droplets and senescence, respectively (See Appendix A for details). Immunofluorescence staining on sections was performed as described for pre-fixed samples in [44] using the following antibody dilutions: chicken anti-L-Plastin at 1:500 (kind gift of P. Martin), secondary antibodies according to species specificity of primary antibodies at 1:500 (Alexa Fluor secondary antibodies, Thermo Fisher Scientific, Waltham, MA, USA). DNA was counterstained with Hoechst. Slides were mounted in ProLong Gold Antifade Mountant (Invitrogen, Thermo Fisher Scientific, Waltham, MA, USA).

### 2.9. Whole-Mount Immunofluorescence Staining and Tissue Clearing

Hearts were fixed in 4% PFA overnight, dehydrated in an increasing gradient of methanol, and incubated in methanol at −20 °C. Whole-mount immunofluorescence staining and tissue clearing using ScaleA2 [45] were performed as previously described [44]. Briefly, whole-mount tissue was permeabilized with 0.3% Triton X-100 in PBS, antigen retrieval was performed by incubation in 65 °C warm 10 mM sodium citrate buffer (pH 6.0) for 30 min, blocked (5% NGS, 1% DMSO, and 0.5% Triton X-100) for at least 2 h at room temperature, and incubated with the respective primary antibodies (diluted in blocking solution) at 4 °C for 3 days. The following dilution were used: chicken anti-L-Plastin at 1:500 (kind gift of P. Martin), mouse anti-Tropomyosin at 1:20 (CH1, Developmental Studies Hybridoma Bank, University of Iowa, Iowa City, IA, USA), rabbit anti-GFP 1:200 (A-11122, Invitrogen, Thermo Fisher Scientific, Waltham, MA, USA), chicken anti-GFP at 1:400 (ab13970, abcam, Cambridge, UK). rabbit anti-PU.1/Spi1B at 1:100 (GTX128266, GeneTex, Irvine, CA, USA), rabbit anti-Fibronectin at 1:100 (F3648, Sigma-Aldrich, St. Louis, MO, USA). Samples were incubated with secondary antibodies according to species specificity of primary antibodies at 1:500 (Alexa Fluor secondary antibodies, Thermo Fisher Scientific, Waltham, MA, USA) and stored in clearing solution at 4 °C until imaging.

### 2.10. Imaging of Fixed and Stained Samples

Images of histological stainings and in situ hybridization were acquired with a slide scanner AxioScan Z1 (Zeiss, Jena, Germany). Fluorescent images of tissue sections were acquired with a Zeiss Axio Imager or Zeiss Axio Observer Z1 equipped with an ApoTome slider for optical sectioning (Zeiss, Jena, Germany). Cleared and whole-mount stained specimen were imaged in ScaleA2 with a light sheet microscope (Lightsheet Z1, Zeiss, Jena, Germany) enabled for dual side illumination and equipped with a 20× detection objective.

### 2.11. Ex Vivo Live Imaging of the Zebrafish Heart

Hearts of old and young transgenic (*mpeg:YFP*) animals were dissected out into PBS, the atrium was removed and the ventricle was mounted into 0.9% agarose (in PBS) into a glass capillary (internal diameter 2.19 mm, outer diameter 2.75 mm) and mounted (one heart at the time) into the sample holder of a light sheet microscope (Lightsheet Z1, Zeiss, Jena, Germany). The hearts were incubated in PBS at 29 °C, which insured cell (mpeg+) activity for more than three h after the dissection of the ventricle. Images were acquired every 2 min for 90 cycles (2 h 58 min) using dual side illumination and a 20× detection objective.

### 2.12. Image Processing, Analysis and Statistics

ZEISS ZEN software (blue addition) was used for brightness and contrast adjustment, dual side fusion (lightsheet data) and image export. Three dimensional (3D) reconstruction and animation was performed with arivis Vision4D. For injury measurement, collagen, fibrin and muscle area segmentation were trained using the machine learning algorithms of the ZEISS software module ZEN Intellesis. Trained models were then applied for image segmentation and analysis to measure wound area and intact muscle area. Evenly-spaced AFOG-stained serial sections of the whole heart were analyzed. Analyses included the ventricular area only. To calculate the percentage of ventricular injured area, the total wound (fibrin+ collagen) area was normalized to the total muscle area for each heart. Collagen amount in uninjured and sham treated hearts was similarly analyzed. Valves were excluded from collagen measurements. Individual models for each staining and time point image set were trained. The software module ZEN Intellesis was also used to train models for L-Plastin counts and muscle area in fluorescent images. The L-Plastin counts per section were normalized to the measured muscle tissue area of the section. Macrophage tracking was performed with arivis Vision4D. Briefly, first the voxel operation denoising was done using particle enhancement, followed by segment generation using the watershed function to identify cells, segment feature filtering for volume to remove debris and the tracking analysis as final segment operation. If necessary, a drift correction was performed before analysis. Segment counts, sphericity, speed and displacement were computed by arivis Vision4D and used for further statistical analyses. Tracks shorter than five time points were excluded from the analysis. In case of a remaining beating movement, the respective time points were excluded from analysis. Movies and images color coded for sphericity were generated with arivis Vision4D. Statistics (one-way ANOVA, post-hoc Tukey HSD, Wilcoxon rank sum test, *t*-test) were computed with R. The boxplots indicate the median of the data. The upper and lower hinges correspond to the first and third quartiles (inner quartile range). The upper and lower whiskers extend from the hinge to the highest and lowest values, respectively (within 1.5 × inner quartile range). Outliers are plotted as points.

## 3. Results

### 3.1. Global Transcriptional Changes in the Aging Zebrafish Ventricle Are Associated with the Actin Filament and the Immune System

In order to identify hallmarks of the aging zebrafish heart, we performed transcriptional profiling using ventricles of 7-month-young and 4-year-old zebrafish (Figure 1A). Differential gene expression analysis identified 1233 genes as differentially regulated with age (DEGs, *p*_adj_ < 0.05) (Figure 1B, Appendix A). Four hundred and eighty-eight genes were significantly down-regulated in old fish compared to young and 745 genes were significantly up-regulated. Quantitative RT-PCR analysis of two of those newly-identified DEGs (*ncf1*, *enpp1*) confirmed the up-regulation in ventricles of 4-year-old fish, whereas relative gene expression levels did not significantly change in 1.5-year-old zebrafish compared to 6-month-old fish (Appendix A).

To assess the relevance of our gene expression dataset, we next compared our 1233 DEGs in old versus young ventricles with genes identified as differentially expressed in the zebrafish *klotho* mutant, a genetic aging model [32]. The hormone alpha-Klotho regulates lifespan in mammals and zebrafish, as respective mutants prematurely age [32,46]. About 42% of the DEGs of our study were also found to be differentially expressed in hearts of *klotho* mutants and 92.7% of those were regulated in the same direction in old versus young and *klotho* mutant versus wild types (Figure 1C,D, Appendix A). This striking overlap suggests that these transcriptional changes can be considered major hallmarks of the aging zebrafish heart. In order to identify processes that might be associated with the changes in gene expression, we performed a gene ontology (GO) enrichment analysis. Biological processes associated with the 488 down-regulated DEGs in the fish ventricle were mainly linked to actin-related terms and muscle components were among the enriched cellular compartments associated to those DEGs (Figure 2A,B, Appendix A). Electron microscopy confirmed that muscle organization is affected in old zebrafish ventricles. While in young muscle tissue, a very regular, compact and evenly spaced sarcomere structure could be observed, muscle tissue from old fish was less uniform and showed more gaps. Mitochondria in old tissue were also interspersed in a more irregular fashion (Figure 2C–E).

The top enriched biological processes associated with the 745 up-regulated DEGs in the fish ventricle were mainly associated with immune system terms (Figure 3A, Appendix A). Strikingly, terms associated with the actin cytoskeleton and immune system were similarly enriched among the terms associated with the 520 DEGs found in the overlap of our dataset and the *klotho* model (Appendix A). Interestingly, numerous chemokines are among the immune system-associated genes that were found to be up-regulated in old ventricles (Figure 3A). Senescent cells, which are a hallmark of aging, are known to secrete high levels of inflammatory cytokines and immune modulators [47]. Surprisingly, among the annotated genes for senescence in zebrafish only *tp53* and *serpine1* (*pai-1*) was upregulated in old ventricles, but not the commonly used marker genes *cdkn1a* and *cdkn2a/b* (*p21* and *p16*, respectively) or *il1b*. (Appendix A). Still, senescence-associated beta-galactosidase staining revealed the presence of senescent cells in the old zebrafish ventricle while only very few senescent cells were detected in the young ventricle (Appendix A). Among the most significantly up-regulated DEGs, genes encoding factors of the complement system—which is a part of the non-cellular innate immune system—were found, such as *c4* and *c6* (Figure 3A, Appendix A). We therefore checked the expression changes of 42 complement components encoded in zebrafish and identified 17 as significantly up-regulated in old ventricles compared to young (Figure 3B). This indicates that the complement system contributes as part of the non-cellular immune system to a state of inflammation in old fish hearts compared to young hearts. Interestingly, the complement system consists of circulating plasma proteins that get locally activated by pathogens leading to a cascade of reactions including the activation of effector molecules. Additionally, the complement system is also involved in the removal of necrotic and apoptotic cells. Notably, our dataset highlighted changes on the gene expression level and not on the protein level, where complement activation occurs.

We addressed the expression of the most significantly changed complement component *c6* further. In situ hybridization revealed expression at the edge of the ventricle close to the atrium, bulbus arteriosus and in areas of pericardial fat (Appendix A). The gene *c6* was expressed in similar domains in young and old fish and no quantitative conclusions can be drawn from this colorimetric expression analyses. The expression of *c6* was mainly observed in non-immune cells in the heart (Appendix A). We thus investigated whether an enriched activation of the immune system is not only reflected by the complement components but whether cell-mediated immunity is also increased in old compared to young hearts. L-Plastin, encoded by the *lcp-1* gene, is a commonly used pan-leukocyte marker in zebrafish and its expression was strongly up-regulated upon age in our dataset (Figure 3C). Immunostaining of heart sections of old and young fish for L-Plastin confirmed an increased presence of immune cells in the old ventricle compared to the young (Figure 3D,E).

### 3.2. Immune Cells of Mainly Myeloid Origin Accumulate in the Heart upon Aging

Whole-mount immunofluorescence, in combination with tissue clearing and light sheet microscopy, revealed that immune cells did not only accumulate but appeared in a more roundish shape and showed the tendency to form clusters on the surface of the ventricle in old fish (Figure 4A,B). In addition to numerous immune cells on the ventricular surface, immune cells were also found to reside intermingled with muscle fibers (Figure 4B, Appendix A). To narrow down which immune cell populations are enriched in the old ventricle, we selected marker genes for different immune cell subtypes in zebrafish or for the differentiation into certain lineages and checked the gene expression regulation in our data set (Figure 4C,D).

The majority of marker genes that was significantly up-regulated was linked to macrophages and their differentiation. Remarkably, only one out of the 46 marker genes, listed in Figure 4D, namely *gata1a*, was significantly down-regulated with age. The up-regulation of *spi1b* together with the down-regulation of *gata1a* in old fish hearts might point towards a skewing from erythroid to myeloid populations as an interplay of *spi1b* and *gata1* are known to determine myeloid versus erythroid cell fate in the developing zebrafish [48]. We employed a second, less biased analysis strategy to identify enriched immune cell population using published single cell RNA sequencing data of the zebrafish whole kidney marrow [49]. We identified unique markers of ten immune cell populations that were distinguished in this previously published study and compared these to our 745 up-regulated DEGs. Of those unique marker genes, 55 were found among our 745 DEGs and 31 out of those 55 were linked to the myeloid lineage. This additional analysis confirmed that the majority of immune cell markers among the up-regulated DEGs is linked to immune cells of myeloid origin (neutrophils, macrophages), mainly macrophages (Appendix A). Still, additional immune cell subtypes were present in the heart, presumably at lower numbers, according to our expression analysis. Among the significantly up-regulated marker genes were *mpeg1.1* (*macrophage expressed 1*, *tandem duplicate 1*), associated with macrophages and a subset of B-cells in the adult zebrafish [50], and *spi1b* (*Spi-1 proto-oncogene b*, also known as *pu.1*), which functions upstream of or within myeloid cell differentiation [35]. We used a reporter line that expresses *YFP* under the control of the *mpeg1.1* promoter to investigate macrophages in the young and old fish heart. Whole-mount immunofluorescence stainings in combination with the endogenous YFP signal of the line showed that mpeg1.1-positive cells were found enriched in old ventricles, especially within immune cell clusters (Figure 4E). These immune cells were sometimes, but not exclusively, found in areas of pericardial fat (Appendix A). Similarly, Spi1b-positive cells were often found within cell clusters of L-Plastin-positive cells in old ventricles (Figure 4F). As the anti-Spi1b antibody did not only stain Spi1b-positive nuclei but also coronary vessels (Appendix A), we also identified coronary vessel-associated clusters (Appendix A). However, not all cells within such an L-Plastin-positive cell cluster in old hearts were mpeg or Spi1B-positive, suggesting that the cell clusters are of a mixed cell identity with a high percentage of myeloid cells. Immunostaining of heart sections of old and young fish did not reveal neutrophils to be increased in numbers in old compared to young but revealed variations between individuals (data not shown). Taken together, these immunostainings show that immune cells of the myeloid lineage, mainly of the macrophage lineage, accumulate in old ventricles and are often found in clusters.

### 3.3. Mpeg-Positive Cells Show Changes in Behavior Linked to Changed Morphology

As actin-cytoskeleton terms, as well as terms linked to locomotion and chemotaxis, were among the enriched biological processes associated with the observed changes in gene expression (Figure 2A and Figure 3A), we decided to further investigate cell locomotion. We thus established an ex vivo live imaging approach to track mpeg-positive immune cells (macrophages, subset B cells) using a reporter line expressing *YFP* under the control of the *mpeg1.1* promoter, followed by a segmentation and tracking analysis (Figure 5A, Appendix A).

Quantification of the tracked cells confirmed increased numbers of mpeg+ cells in old fish hearts compared to young as suggested by RNA-Seq data and immunofluorescence stainings (Appendix A). Not only did cell numbers increase but also an increased abundance of roundish cells was observed when color-coding for sphericity (Figure 5B). The analysis of cell sphericity of all tracked cells further demonstrated that mpeg+ cells have a more roundish shape in old compared to young hearts as observed for pan-immune cells (L-Plastin+ cells) in immunofluorescence analysis (Figure 4 and Figure 5C). The average track speed was increased in the old hearts indicating an increased cell migration (Figure 5D). Directed cell movement was neither observed in young nor in old hearts. We therefore compared the mean squared displacement as a common measure of random movement. The mean squared displacement was increased in the old fish hearts, too (Figure 5E). When assessing cell sphericity versus cell speed, a clear shift of cell population can be observed in old compared to young (Figure 5F). A general shift of the cell population in old to the right and thus towards more roundish cells can be observed, in line with the sphericity quantification (Figure 5C). Some of these more roundish cells in the old display a higher speed as revealed by the upward shift of the population compared to the young cell population. At the same time, we also observe a considerable part of the cell population in old as more roundish but less mobile cells as revealed by the dense lines in the lower right corner. Thus, the distribution of cell speed is more spread in the old than in the young. Taken together, these data suggest a changed immune cell locomotion in old versus young fish hearts.

### 3.4. Wound Size, Regeneration Response and Collagen Deposition Changes upon Cryoinjury in Old Compared to Young Hearts

Given the outstanding regenerative capacity after myocardial injury and a known role of the immune system in regeneration, we next asked whether regenerative potential might be impaired in aged zebrafish hearts. We injured hearts of old zebrafish in comparison to young zebrafish by cryoinjury and measured the ratio of wound to intact muscle 30 days post injury (dpi). Acid fuchsin orange G (AFOG) staining visualized a remaining scar consisting of collagen and fibrin in young and old fish (Figure 6A).

While scar size was quite uniform in young hearts, striking differences in fibrotic scar appearance were observed within the old fish population. While some old fish hearts had regenerated similarly to young, some ventricles (57%) still showed a prominent scar. These scars sometimes were not yet covered by a thickened myocardial layer and some were characterized by a large amount of collagen that lined the presumably entire injured area in the most severe cases. While the median scar percentage was below 2% in young at 30 dpi, it was in approximately 7% in old and fibrotic tissue measured more than 40% in the most severe case (Figure 6B). Whole-mount immunostainings for the muscle marker tropomyosin (TPM) at 30 dpi confirmed the persistence of significant areas in old, injured hearts that were devoid of functional muscle tissue (4 out of 7 old hearts) (Appendix A). The collagen amount in the remaining scar was significantly increased in scars of the old fish cohort (Figure 6C). These significant amounts of collagen at 30 dpi could point either towards an impaired scar resolution upon aging or to an increased scar deposition in the old. In order to address the latter, we analyzed the fibrotic scar in young and old fish at 7dpi. The collagen amount in the scar was already significantly increased at 7dpi in old fish hearts compared to young hearts (Figure 6D). In young hearts, collagen deposition indicated by blue staining was mainly observed at the border zone between injured area (blue-red area) and intact muscle (in orange) (Figure 6E). In addition, increasing collagen amounts were observed along the epicard at the injury site in the young hearts. This collagen layer lining the outside of the injured area showed a thickened appearance in old ventricles. The injured tissue and forming scar showed differences in structure and composition. The injured tissue was not as compact as in young and appeared loose towards the intact muscle. Collagen deposition was also observed in scars of old fish with increasing amounts towards the border zone. Such differences in the structure and composition of the injured heart tissue were already observed in a subset of fish (2 out of 5) at 3 dpi (Figure 6F). The injured area was less compact and stained in a darker red. Three out of five old fish displayed injured areas that were comparable to those in young animals. These data suggest that the initial response to injury including scar formation is different in at least a subpopulation of old and young fish. Interestingly, collagen accumulation was also increased in old uninjured hearts as well as sham treated control hearts at 30 dpi (Appendix A). None of the recently identified marker genes for macrophage subtypes involved in scarring or scar remodeling, namely *tnfa*, *spp1* and *csf1ra*, (Bevan et al., 2020), were found to be significantly changed during homeostasis with age in our dataset (Appendix A). Taken together, these data show that the regenerative response to cardiac injury is impaired in a subset of old animals while some zebrafish still show significant regenerative capacities at an age of about 4 years.

## 4. Discussion

Zebrafish possess outstanding regenerative capacities and are increasingly used as a model for research on aging. Here, we investigated transcriptional changes in the aging zebrafish heart. We identified the immune system as activated in the old fish and found muscle organization to deteriorate upon aging. Strikingly, the enrichment of immune system and actin-associated terms was not only identified in our set of differentially expressed genes but also when comparing it to expression data from a genetic aging model, namely *klotho* zebrafish [32]. This points towards the activation of immune system-associated genes upon aging in the zebrafish heart. A state of increased inflammation in the elderly is commonly referred to as inflammaging. In the context of the senescence-associated secretory phenotype (SASP), considered a hallmark of aging, high levels of inflammatory cytokines and immune modulators are secreted by senescent cells (reviewed in [51]). However, senescent cells and SASP might only play a minor role in the observed up-regulation of the immune system in the fish heart upon aging since senescence-associated marker genes such as *cdkn1a* and *cdkn2a/b* were not significantly changed. This is thus different than has recently been described in mice where senescent cells accumulate in adipose tissue and liver upon aging and induce macrophages to proliferate [52]. Instead, genes encoding components of the complement system, that is, part of the non-cellular immune system, were significantly upregulated. The complement system is a complex network of plasma and membrane-associated serum proteins and its activation is organized into a hierarchy of proteolytic cascades. Gene expression might therefore not directly reflect the activation state of the complement cascade in the heart. The number and activation of complement genes, especially in the aging fish heart, is striking though. Complement proteins are suggested to contribute to myocardial damage in humans and are found up-regulated and activated in the human heart after myocardial infarction [53]. The observed activation of the complement system in our study might therefore reflect an increased damage in the aged zebrafish heart. C3b for opsonization of debris and complement receptors on macrophages are both needed for debris removal. We find activation of *c3* genes as well as the receptor (*C3R/itgam*). Whether this activation is detrimental, leading to tissue destruction, or beneficial for tissue clearance needs to be clarified. Several complement inhibitors are available and can limit cardiac damage (reviewed in [54]). Interestingly, the complement system factor C5aR1 has been identified as important for successful cardiac regeneration zebrafish, axolotls, and mice [55]. However, expression of *c5ar1* was not changed in our dataset. In conclusion, the activated immune system, including the complement system, as observed here in the old zebrafish heart points towards an “inflammaging”-like state.

In addition to humoral components of the innate immune system, we identified cellular components of the immune system as increased in the ventricle with age. Numerous immune cells reside on the ventricular surface or intermingled with muscle fibers. The major enriched cell types are of myeloid origin. Furthermore, we show that macrophage morphology and behavior changes in the old compared to the young fish heart. Specifically, macrophages became more spherical and motile. Interestingly, a similar observation has been made in zebrafish for microglia, which are the macrophages of the brain. Microglia cells in the adult zebrafish telencephalon display larger cell bodies and shorter processes with aging [56]. As macrophages are phagocytic active cells, resident macrophages in the zebrafish heart are likely to play a role in removing debris. In mice, a recent study identified macrophages present within the healthy myocardium and taking up material derived from cardiomyocytes, including mitochondrial remnants [57]. Upon aging, the amount of debris is supposedly increased, resulting in an elevated need for phagocytosing cells. The complement system with its known chemotactic anaphylatoxins C3a and C5a might serve as a link as increased gene expression of the complement receptor encoding genes *cr3* (*itgam*) and *c3ar1* was observed in the old fish heart. This might reflect the increased need for neutrophils and macrophages, potentially at least partially recruited through the complement. In conclusion, the activated immune system as observed here in the old zebrafish heart, might be a conserved age-associated phenomenon. This underlines the potential to investigate zebrafish to further understand processes in the aging heart. Finding the balance between sufficient activation of the immune system to help tissue clearance and a tight regulation to reduce inflammation and protect heart tissue might be key for healthy organ aging.

In contrast to a previous study [20], we observed impaired regenerative capacities manifesting itself in an increased wound size after injury in old zebrafish. These are not necessarily conflicting results, since first the age group of old animals used in the previous study was younger than the fish used in our study and second the previous study used ventricular apex resection as injury model, whereas we performed cryoinjury. We consider our aging cohort of fish to represent a truly old fish as opposed to those used in the study reporting life-long regenerative capacity, given a reported median life span of approximately 36 to 42 months and a maximum life span of up to 66 months for zebrafish [23]. Furthermore, ventricular resection is based on tissue removal and therefore induces less necrosis and scar formation than the tissue damage induced by cryoinjury. Debris clearance is therefore not required to the same extent after resection. Our study showed increased collagen accumulation in response to injury indicating an increased scarring response with age. This difference in local necrosis and scar formation might thus explain the different regeneration outcome, at least partially.

Interestingly, we observed a wide range of wound sizes at 30 dpi in old compared to young ventricle. While a subset of old fish had a reduced scar size similar to young animals, a subset of old animals (about 60%) showed increased scar sizes and reduced muscle fibers covering the injured area. Since the scar was not only larger at 30 dpi but collagen accumulation was also increased in wounds at already 7 dpi and wound morphology was different at 7 and 3 dpi in a subset of old animals, we conclude that the observed differences in the age groups is due to differences in the initial injury response and scarring rather than resorption of the scar tissue. Immune cells are known to be important for many regenerative processes in several organ and model systems such as salamander limb, the neonatal mouse heart, the zebrafish and medaka heart, and the zebrafish spinal cord or retina [58,59,60,61,62]. Macrophages are directly linked to the cardiac scarring process [63]. It is therefore of interest to further understand to what extent the increased presence of immune cells might modulate the regenerative responses after cardiac injury in old fish. A recent study identified specific macrophage populations involved in scar deposition and subsequent resolution during zebrafish heart regeneration [64]. This previous study proposed *tnfa* as a marker for early Collagen I-scarring macrophages, *csf1ra* as important in promoting pro-inflammatory macrophage-mediated scar deposition and *spp1* as involved in initial scar deposition as well as later-stage collagen remodeling. In our dataset, neither *tnfa* nor *csf1ra* nor *spp1* were found to be significantly changed during homeostasis with age. However, since we did not investigate the presence of these cell types during regeneration, we cannot rule out that a change of these macrophage subtypes is involved in the differences in scarring during regeneration. Still, no difference in such macrophages has been observed during homeostasis that could be responsible for the difference in the injury response.

It has been hypothesized that immune system specification comes at the expense of regeneration and the decision between tissue regeneration or scarring is determined by a variety of factors (reviewed in [65]). The tight balance between this scarring and a regeneration scenario might be disturbed in the subset of old animals that show an impaired regenerative response. The up-regulation of the immune system in the old might thus be detrimental to the regeneration response.

Alternatively, the regeneration capacity could be reduced due to reduced de-differentiation and cycling capacities of muscle cells that are needed to rebuild the missing muscle structures. Such observations have been made for zebrafish that lack telomerase activity [66]. Zebrafish deficient in telomerase show premature aging and a reduced life span [67,68]. Since telomere shortening is a hallmark of aging, impaired telomerase activity with age might be a likely mechanism to influence regeneration capacities in zebrafish in addition to the immune system. Telomere length as well as the inflammatory state of individual fish are likely to differ to some extent and might cause such differences in the regeneration capacity in aged zebrafish.

Taken together, our study indicates the presence of an ‘inflammaging’-like process in the zebrafish heart that includes increased complement gene expression and increased numbers of resident immune cells with a yet to be determined impact on the regenerative capacity of the heart in old animals.

## Figures and Tables

**Figure 1 cells-11-00345-f001:**
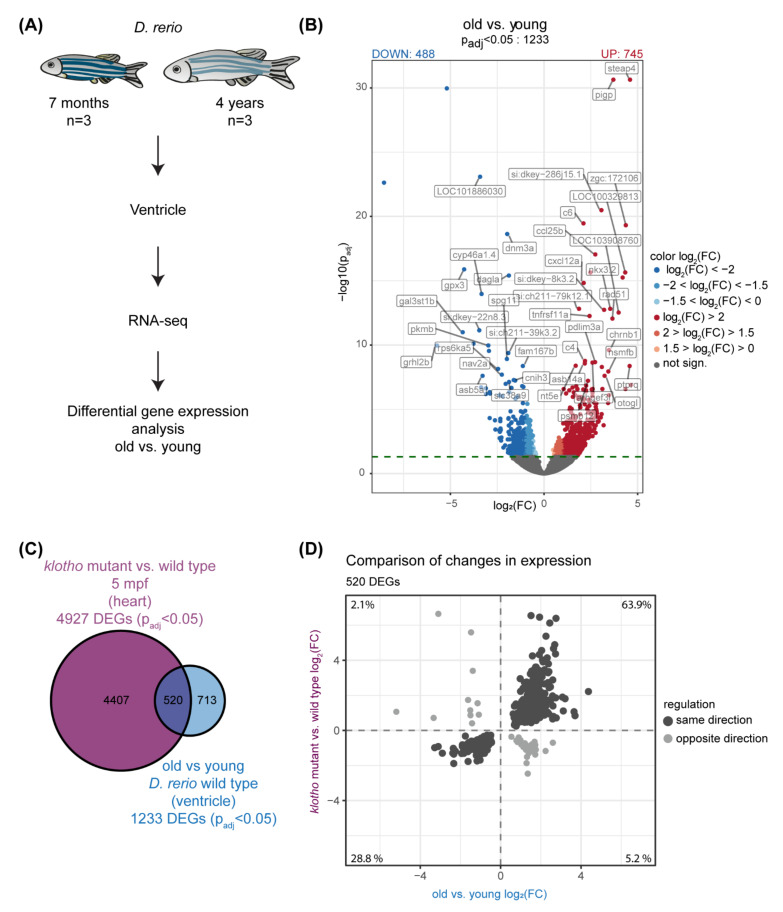
Transcriptional analysis identifies global changes in the aging zebrafish heart. (**A**) Scheme illustrating our transcriptome profiling approach with ventricles of 7-months (young) and 4-year (old) old zebrafish. *n* = 3. (**B**) Volcano plot illustrating differential gene expression (DEG) analysis that identified 1233 differentially expressed genes (DEGs); *p*_adj_ < 0.05 (dotted line). DEGs down-regulated in old vs. young are highlighted in blue (488), DEGs up-regulated in red (745). Gene symbols of top 50 DEGs (most significantly changed) are shown. FC: fold change. (**C**) Comparison DEGs in old vs. young with DEGs in hearts of *klotho* mutant vs. wild type [32] identifies 520 overlapping genes (**D**) 92.7% of 520 overlapping genes (in dark grey) are regulated in the same direction in old vs. young and mutant vs. wild type.

**Figure 2 cells-11-00345-f002:**
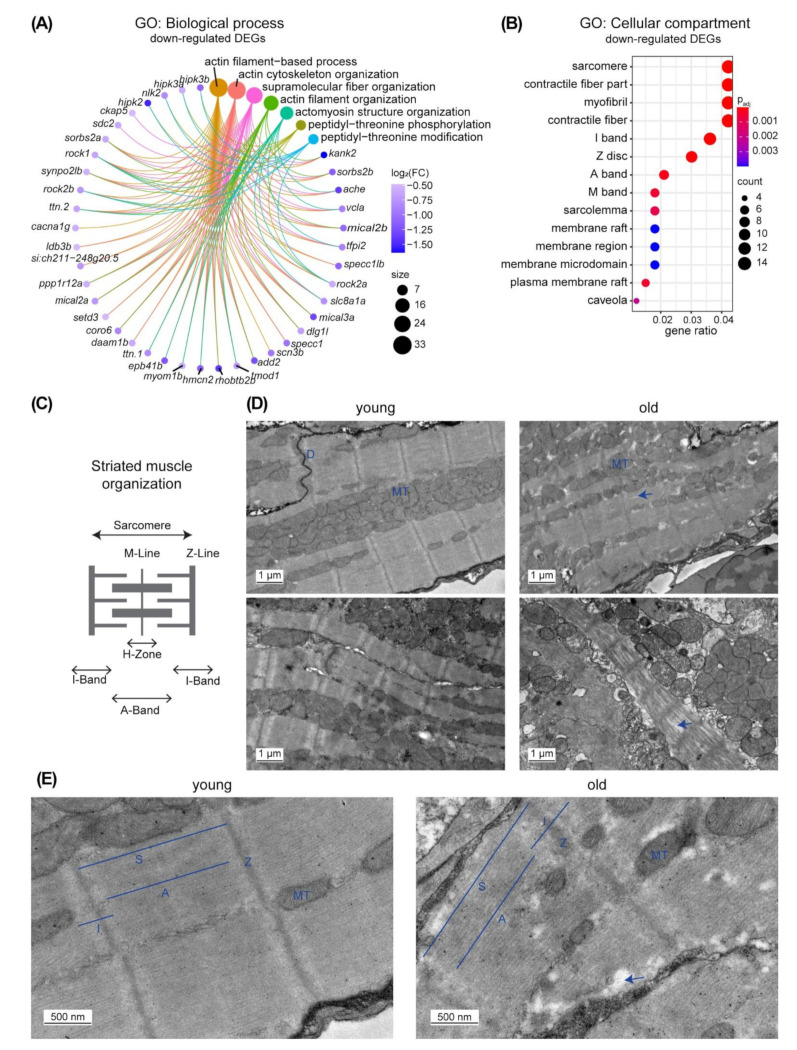
Expression of genes related to the actin filament is changed upon aging. (**A**) Circular plot illustrates the GO terms of the category “biological process” identified as enriched among 488 down regulated DEGs and the respective genes associated with these terms. The size of the GO term circle indicates the number of the genes and the color of the gene circle indicates the fold change (log_2_(FC)) in blue shades. Actin-related terms are prominent. (**B**) Dot plot indicates GO terms of the category “cellular compartments” that are enriched among the 488 down regulated DEGs. The size of the circle indicates the number of the genes, the color the adjusted *p*-value of the GO term analysis. Terms related to muscle structures are prominent. (**C**) Scheme of the striated muscle organization. (**D**) Electron microscopy pictures of the ventricle of 1-year-old (young) and 5-year-old (old) fish indicate changes in muscle organization in old. Scale bars: 1 µm. (**E**) Higher magnification picture of striated muscle organization in young and old. Scale bars: 500 nm. D: Desmosome; MT: Mitochondria; S: Sarcomere; I: I-Band; A: A-Band; Z: Z-Line. Arrows indicate to less dense muscle structures in old.

**Figure 3 cells-11-00345-f003:**
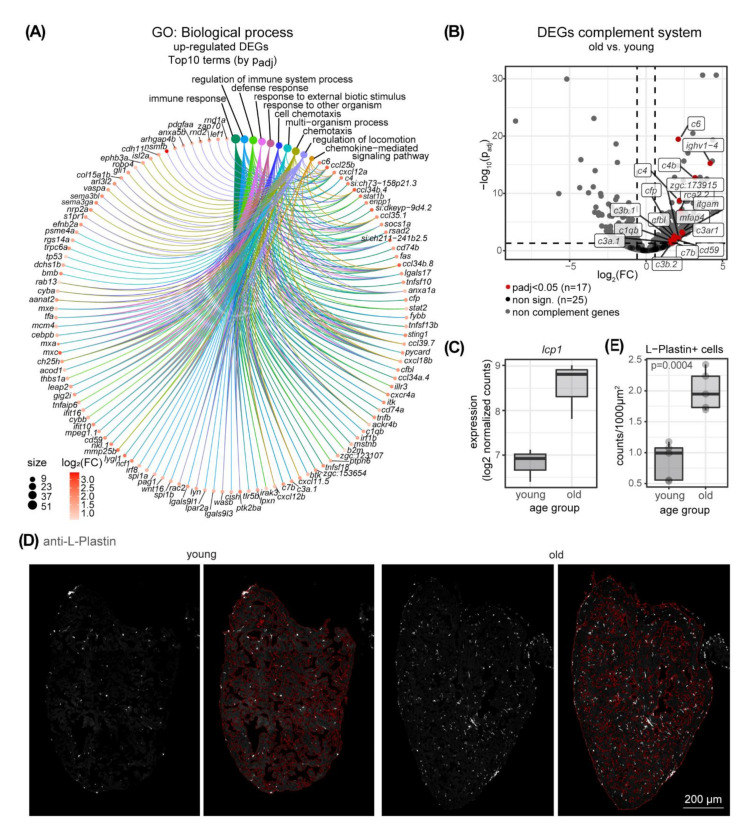
Expression of genes related to the immune system is changed upon aging. (**A**) Circular plot highlights the top 10 (by adjusted *p*-value) GO terms of the category “biological process” identified as enriched among 745 upregulated DEGs and the respective genes associated to these terms. The size of the GO term circle indicates the number of the genes and the color of the gene circle indicates the fold change (log_2_(FC)) in red shades. Immune system-related terms are prominent. (**B**) DEGs encoding components of the complement system are highlighted FC: fold change (**C**) Expression of the pan leukocyte marker gene *lcp1* as log2 of normalized counts in young and old ventricles. *n* = 3. (**D**) Immunostaining of heart sections of old (37 months) and young (8 months) fish for the pan-leukocyte marker L-Plastin. The muscle tissue area that was segmented using ZEN Intellesis, is indicated in red in the respective right panels. Scale bar is 200 µm. (**E**) Quantification of L-Plastin-positive cells (from (**D**)) as counts per 1000 µm^2^ muscle area. *n* = 5; two-sample *t*-test *p* = 0.0004.

**Figure 4 cells-11-00345-f004:**
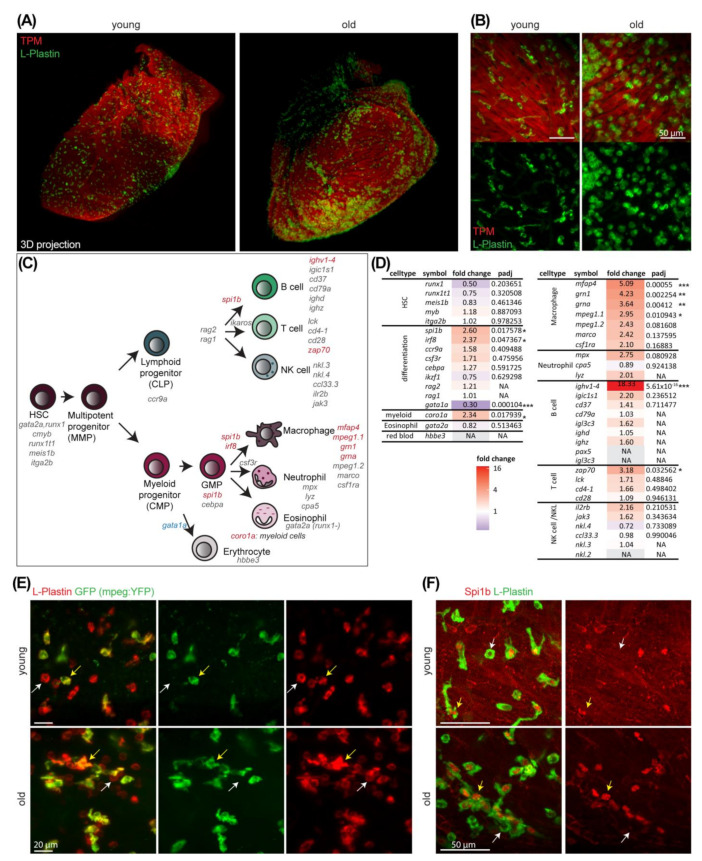
Immune cells accumulate in ventricles upon aging. (**A**) Whole-mount immunostaining of old (4 years) and young (1 year) fish hearts for the pan-leukocyte marker L-Plastin (green) and the muscle marker Tropomyosin (TPM) (red) are shown. Lightsheet microscopy and 3D projection reveal increased cell numbers and cell cluster formation in old hearts. (**B**) Higher magnification images of old and young hearts stained with L-Plastin (green) and TPM (red) reveals that immune cell numbers increase and cell morphology changes to be more roundish in old hearts. Scale bar: 50 µm. Whole-mount immunostaining and lightsheet microscopy. (**C**) Scheme of different immune cell types and their respective lineages in zebrafish. Known marker genes for different cell populations as well as for differentiation into specialized cell types are listed. Genes with significantly up or down regulated expression in old are highlighted in red or blue, respectively. (**D**) Immune cell markers (from (**C**)), their respective fold change with age and adjusted *p*-value (*p*_adj_) are listed (transcriptional profiling from Figure 1, *n* = 3). Fold change values are color-coded with up in red, no change white and down in blue. NA: not available. *: *p*_adj_ < 0.05; **: *p*_adj_ < 0.01, ***: *p*_adj_ < 0.001 (**E**) Whole-mount immunostaining of young (1 year) and old (3 years) fish hearts of a *mpeg:YFP* reporter line for the pan-leukocyte marker L-Plastin (red) and GFP (green) reveals that a large subset of L-Plastin+ cells are mpeg-positive. White arrow indicates a single positive immune cell and yellow arrow indicates a double positive cell. Lightsheet microscopy. Scale bars: 20 µm. (**F**) Whole-mount immunostaining of young (1 year) and old (4 years) fish hearts for the pan-leukocyte marker L-Plastin (green) and Spi1b (red) reveal large subset of L-Plastin cells are Spi1b-positive. White arrow indicates a single positive immune cell and yellow arrow indicates a double positive cell. Lightsheet microscopy. Scale bars: 50 µm.

**Figure 5 cells-11-00345-f005:**
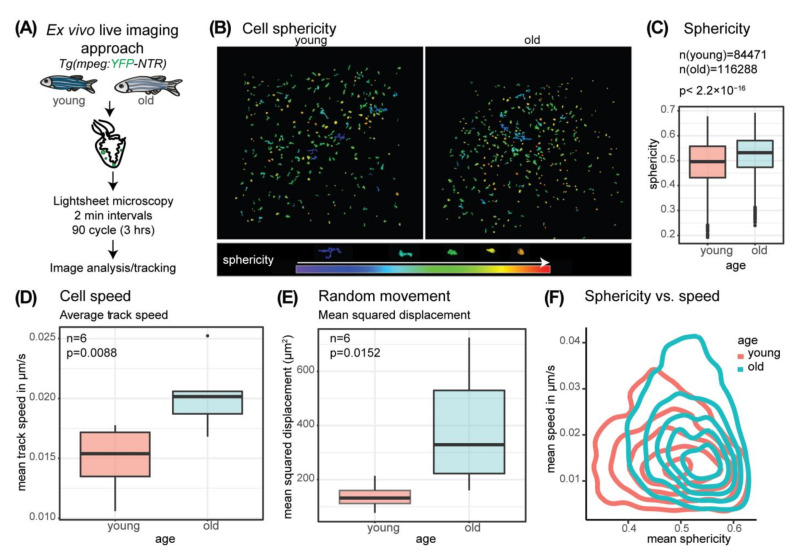
Macrophage morphology and behavior changes with age. (**A**) Schematic overview of the ex vivo live imaging approach using a reporter line expressing YFP under the control of the *mpeg1.1* promoter (**B**) Sphericity of segmented and tracked objects (mpeg+ cells) at timepoint (tp) = 1 is color coded with red as high sphericity (round) and purple/blue as low sphericity. Age young: 12 months, age old: 4 years 7 months. The legend includes a color code for sphericity and representative example cell shapes. (**C**) Comparison of sphericity of all tracked objects (mpeg+ cells) at all time points analyzed in old and young fish hearts reveals a significant shift to higher average cell sphericity in the old. Welch two-sample *t*-test *p* < 2.2 × 10^−16^, age young: 3 × 6 months and 3 × 12 months, age old: 3 × 4 years 7 months and 3 × 5 years. (**D**) The boxplot illustrates the mean track speed of all segmented tracks analyzed per biological replicate. 6 old and 6 young hearts were analyzed. Quantification of average speed of mpeg+ cells in old and young fish ventricles indicate a faster cell movement in old. *n* = 6; two-sample *t*-test *p* = 0.0088, age young: 3 × 6 months and 3 × 12 months, age old: 3 × 4 years 7 months and 3 × 5 years. (**E**) The boxplot illustrates the average mean squared displacement analyzed per heart of 6 old and 6 young hearts. Quantification of mean squared displacement of mpeg+ cells in old and young ventricles reveals a higher random movement in old. *n* = 6, Wilcoxon rank sum test *p* = 0.0152, age young: 3 × 6 months and 3 × 12 months, age old: 3 × 4 years 7 months and 3 × 5 years. (**F**) Density plot showing the average cell sphericity per track vs. average speed per track reveals a shift of cell populations in old towards more roundish and to faster cells.

**Figure 6 cells-11-00345-f006:**
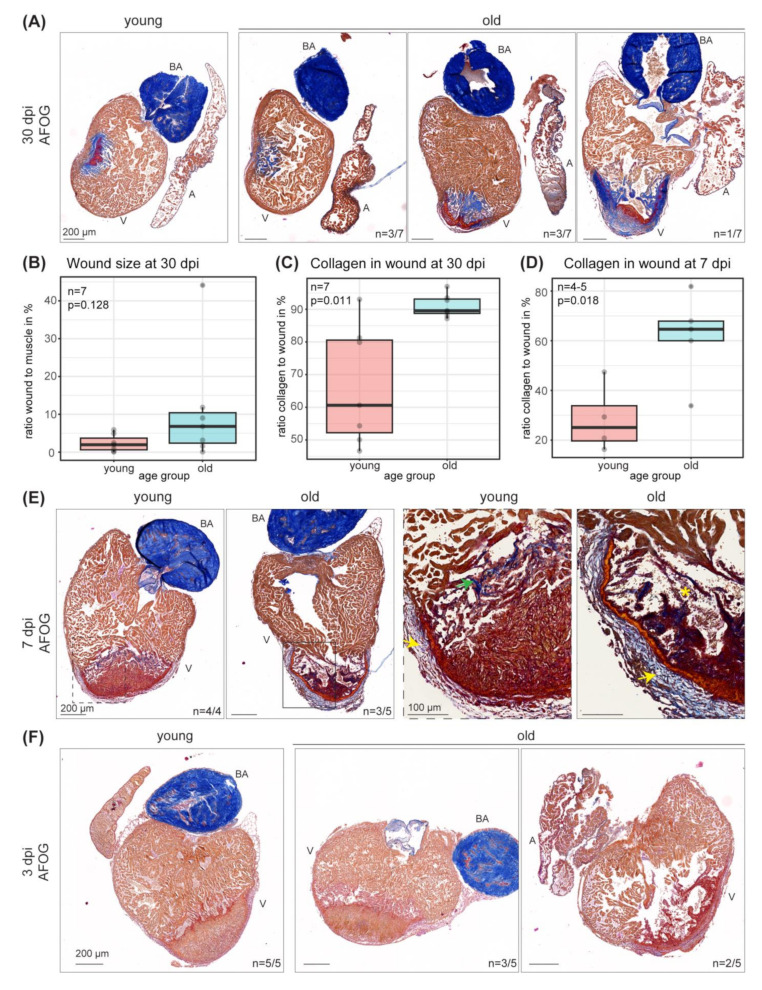
Wound size, regeneration response and collagen deposition changes upon cryoinjury in old compared to young hearts. (**A**) Acid fuchsin orange G (AFOG) staining of representative sections of young and old hearts 30 dpi visualizes the scar area in red (fibrin) and blue (collagen). Intact muscle is stained in orange. Scars display strong differences within the old (3 years 10 months) population as compared to young (1 year 4 months). 4 out of 7 old fish show differences in the injury response compared to young. Scale bars: 200 µm. (**B**) Quantification of the wound as percentage of intact muscle at 30 dpi (from (**A**)). *n* = 7. Wilcoxon rank sum test *p* = 0.128. (**C**) Quantification of collagen in wound (in %) at 30 dpi (from A). *n* = 7. Wilcoxon rank sum test *p* = 0.0111. (**D**) Quantification of collagen in wounds (in %) at 7 dpi (from (**E**)). *n* (young) = 4; *n* (old) = 5; two-sample *t*-test *p* = 0.0177. (**E**) AFOG staining of sections of young (1 year 2 months) and old (4 years 4 months) hearts 7 dpi visualizes the scar area. Scars display differences in young and old (3 out of 5 old). Yellow arrow points towards thickened collagen layer in the region of the epicard. Green arrow points towards collagen accumulation at the wound border in young. Yellow asterisk marks loosened muscle structure. Scale bars: 200 µm in overview and 100 µm in zoom-in. (**F**) AFOG staining of sections of young (9 months) and old (3 years 2 months) hearts 3 dpi visualizes the injured area. A subset of old animals (2 out of 5) show differences in the injury response. Scale bars: 200 µm. A: atrium; BA: bulbus arteriosus; V: ventricle. dpi: days post cryoinjury.

## Data Availability

RNA-seq data generated and discussed in this study have been deposited in NCBI’s Gene Expression Omnibus [69] and are accessible through GEO Series accession number GSE182979 (https://www.ncbi.nlm.nih.gov/geo/query/acc.cgi?acc=GSE182979 (accessed on 30 October 2021)).

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
