# Peer review of "Aging Activates the Immune System and Alters the Regenerative Capacity in the Zebrafish Heart"

_cells, 2022, doi:10.3390/cells11030345_

Round 1

Reviewer 1 Report

In this manuscript authors compared the ventricular transcriptome and the regenerative capacity of old vs. young zebrafish hearts following cryoinjury. Manuscript is well written, subject is of substantial scientific interest and obtained results are important contribution to evolutionary aspects of aging and regeneration of injured heart. 

What needs to be improved is graphical quality of all figures, now they are blur, not of publishing quality and need to improve resolution. Also, authors should comment on Figure 3. (D) showing immunostaining of heart sections of old and young fish for the pan-leukocyte marker L-Plastin where DNA counterstained with Hoechst (blue) is much less abundant in young (left) vs. old (right) image. Marker for L-Plastin in accompanying images are proportionally abundant and it is not clear if this is simply effect of “abundant” tissue or genuine increase with aging. This is my only serious concern regarding this manuscript.

Reviewer 2 Report

Reuter et al. studied how aging activates the immune system and alters the regenerative capacity in the zebrafish heart model system. In the beginning, the authors showed global transcriptional changes in the aging zebrafish ventricle and their association with the actin filament and the immune system. They also showed that the immune cells that accumulate during aging are mainly of myeloid origin, and there is changed morphology in these aged cells. They later revealed that in response to cryoinjury young hearts demonstrate a changed response to wound size, regeneration response, and collagen deposition compared to old hearts. The study is innovative and insightful. It certainly adds a  lot of value to the field. A few grammatical edits are needed, but authors will undoubtedly get it proofread by experts. I have some comments on other aspects that will help in further refinement of the current manuscript. 

  1. Most of the figures in the manuscript are in low resolution and difficult to read.  It would be better if the authors add high-resolution images. 
  2. On page 10, lines 315-320, the Authors mentioned that they see senescence-associated beta-galactosidase, which marks the senescent cells in the heart. Still, they do not see any upregulation of p16 or p21with RNA seq analysis. Did they check other markers for senescence in this scenario, such as IL-6, IL-1B, MMP-3, PAI-1, GDF-15, etc.? At times there are issues with detecting p21 and p16 with RNA seq analysis. It would be helpful if authors use other techniques such as qRT-PCR to detect the levels of the SASP ( senescence-associated secretory phenotype) genes.
  3. The authors showed an increased macrophage number in aged zebrafish but they do not explore the mechanism in detail. In the mouse model, Covarrubias et al. has shown (Nature Metabolism volume 2, pages 1265–1283, 2020) that the macrophage number increases during aging. The main reason behind this proliferation of macrophages is the increased accumulation of senescent cells during aging, and these senescent cells secrete cytokines and chemokines that trigger the proliferation of macrophages. Can authors check if this is the case in zebrafish? This will certainly make the connection with increased immune system cells with aging, as this is still an underexplored area of research.

Reviewer 3 Report

In this manuscript, Reuter et al. generated the transcriptome profiles using the ventricles from young (7 months) and aged (4 years) zebrafish. By further analyzing along with other published data sets, the authors highlighted the enrichment of the different expression genes associated with muscle organization (down-regulated in aged ventricles) and immune response (up-regulated in aged ventricles). Using electron microscopy, live imaging, and immunofluorescence staining, the author confirmed that the muscle organization was impaired in old hearts, and identified that the myeloid lineage cells were enriched in old hearts with more roundish shapes and increasing motility when compared with the young counterparts. Finally, the authors compared the capacity of heart regeneration between the young and old hearts using the well-established cryoinjury zebrafish model. They found collagen accumulation was increased in injured old hearts, although variation was detected, and the collagen accumulation was also found in uninjured old hearts. Overall, the finding in this study is interesting and significant. However, there are concerns on experimental designs and result interpretations. Some data was not convincing at current status. These issues should be addressed before the manuscript is published.

Major comments:

  1. Aged zebrafish are prone to parasitic infection, leading to inflammatory infiltration and immune response in multiple organs. Therefore, the authors should make some efforts in showing the animals (at least the hearts) are free of parasitic infection and healthy.
  2. The description of some figure panels in the text are hard to follow. The author failed to describe the results in essential detail. For example, line 295-297, what are the differences between young and old heart muscle in terms of sarcomere, m-line, z-line, etc that shown in Fig2C? Also, proper qualification is needed to conclude that “muscle structures were less regular and dense in the old ventricle”. Along with this, line 483-492, where is “the border zone between injured area and intact muscle” shown in Fig6E? What phenotype dose the author expect the reader to see in Fig6F? The author should do a self-check and write the results more descriptive and accurate.
  3. It is hard to see any difference between “young” and “old” in Fig5B, instead the dots with different colors. Also, Fig5B was not described. The author should make representative images showing the cell shapes. Line 400-441, the author should define “directed cell movement” in young heart and “squared displacement” in old heart by showing the trajectory of cell migration in the figure. So far, the data shown in Fig5D and Fig5E is not convincing.

Minor comments:

  1. L-plastin is supposed to express in the cytosol. Please explain its expression mainly on the cell membrane in movie1 and movie2.
  2. Lcp-1 is missed in the list of Fig4D.
  3. Please clarify the “46 maker genes” in line 381.
  4. Please add the number of cells analyzed in Fig5D and E.

Round 2

Reviewer 2 Report

The authors have responded to my concerns, but none of them are satisfactory enough. The figures are certainly better now and have high resolution. The authors have added a few more senescence markers, but it is not enough to support the claim. I still feel the manuscript needs more work and more data to support their claims.

Reviewer 3 Report

The authors have answered most of the concerns I raised in the previous version of the manuscript. However, several issues still remain.

  1. I did not see the “zoom-in views as examples of such shape examples” that the authors mentioned in response to my major comment 3. Please incorporate it in the current version of the manuscript.
  2. Please rephrase the description Fig5F (line 458-461). Dose the authors mean two cell groups were shifted from “young” to “old”? (one group is not round with less motility in “young” that became more roundish in “old”; the other group is round with high motility in “young” that became even more mobile in “old”?). So far, the description is confusing to me to interpret the result.
  3. Please label the location of collagen in the zoom-in image of “young” in Fig 6E to match the description in the text “in addition, increasing collagen amounts were observed along the epicard at the injury side in young” (line 502-503).

Apart from these, the manuscript is acceptable for publication.
